# Quantum Transport of Dirac Fermions in HgTe Gapless Quantum Wells

**DOI:** 10.3390/nano12122047

**Published:** 2022-06-14

**Authors:** Gennady M. Gusev, Alexander D. Levin, Dmitry A. Kozlov, Ze D. Kvon, Nikolay N. Mikhailov

**Affiliations:** 1Instituto de Física, Universidade de São Paulo, Sao Paulo 135960-170, Brazil; alexander.d.levin@gmail.com; 2Institute of Semiconductor Physics, 630090 Novosibirsk, Russia; dimko@isp.nsc.ru (D.A.K.); kvon@isp.nsc.ru (Z.D.K.); mikhailov@isp.nsc.ru (N.N.M.); 3Novosibirsk State University, 630090 Novosibirsk, Russia

**Keywords:** quantum transport, HgTe quantum well, Landau levels

## Abstract

We study the transport properties of HgTe quantum wells with critical well thickness, where the band gap is closed and the low energy spectrum is described by a single Dirac cone. In this work, we examined both macroscopic and micron-sized (mesoscopic) samples. In micron-sized samples, we observe a magnetic-field-induced quantized resistance (~*h*/2*e*) at Landau filling factor ν=0, corresponding to the formation of helical edge states centered at the charge neutrality point (CNP). In macroscopic samples, the resistance near a zero Landau level (LL) reveals strong oscillations, which we attribute to scattering between the edge ν=0 state and bulk ν≠0 hole LL. We provide a model taking an empirical approach to construct a LL diagram based on a reservoir scenario, formed by the heavy holes.

## 1. Introduction

The gapless helical edge states flowing along the edge of the two-dimensional (2D) system attract the attention of many due to both fundamental and practical motivations. First, their existence serves as a signature for 2D topological insulators [1,2,3,4,5,6]. Second, the one-dimensional nature of the edge states offers the possibility to study strongly correlated fermion systems, such as the helical Tomonaga–Luttinger liquid [7]. Moreover, the helical edge states can be used to produce Majorana or parafermion modes for quantum computation [8].

Helical edge states arise at the edges of the topological insulator (quantum spin Hall effect) in the absence of an external magnetic field. Particularly, the HgTe-based quantum well with inverted band spectrum can host topological helical states [9,10,11,12,13]. It is expected that these helical channels lead to quantized conductance with the value of $2e^{2}/h$ and nonlocal edge transport [11,13], which has been observed only for short distances between the measurement probes in the range of the few micrometers. The deviation between the theoretical prediction and experimental value has been attributed to many different effects, including effects of Rashba spin–orbit coupling [14,15], charge puddles [16,17] and other numerous sources of inelastic scattering [18].

In addition to the insulator with a bulk gap, helical states can also exist in a gapless system. A remarkable example is 2D massless Dirac fermions in the presence of a strong perpendicular magnetic field, such as graphene [19,20] and gapless HgTe quantum wells [21,22,23,24,25]. It has been demonstrated that at the critical HgTe well thickness $d_{c}$ equal to, depending on the surface orientation and the quantum well deformation, 6.3–6.5 nm, the band gap collapses, and single-valley Dirac fermions can be realized.

In the presence of a strong perpendicular magnetic field, the zero Landau level of the Dirac fermions forms two counter propagating edge states similar to 2D topological insulators [25]. As a result, conductance is zero in the QHE regime and quantized in universal units $2e^{2}/h$ in the quantum Hall (QH)-metal regime in the absence of backscattering between spin-polarized states.

In the present work, we studied the quantum transport in both mesoscopic and macroscopic devices fabricated from HgTe zero-gap quantum structures. In the mesoscopic samples, we observed a magnetic-field-induced, quantized resistance at ν=0. These experiments clearly demonstrate the existence of a robust helical edge state in a system with single-valley Dirac cone materials. In the macroscopic sample, the resistance strongly deviates from the quantized value.

Moreover, we observed large oscillations of the resistance at ν=0. We attribute these oscillations to the elastic intersubband scattering between the edge ν=0 state and bulk ν≠0 hole LL. We observed an unconventional LL diagram for hole Dirac particles with several ring-like patterns, which is attributed to the LL crossing of single LL and manifold-degenerate subband levels. We report a model considering the reservoir of the sideband hole states. The model reproduces some of the key features of the data, in particular, the density dependence of the hole LL and manifold LL crossing points. We propose that this model provides a framework for more sophisticated theoretical tools to understand many-body phenomena, such as spin-splitting enhancement effects.

## 2. Materials and Methods

Quantum wells ${\mathrm{Cd}}_{0.65}{\mathrm{Hg}}_{0.35}\mathrm{Te}/\mathrm{HgTe}/{\mathrm{Cd}}_{0.65}{\mathrm{Hg}}_{0.35}\mathrm{Te}$ with (013) surface orientations and a well thickness of 6.3–6.4 nm were prepared using molecular beam epitaxy (Figure 1a). Two different types of devices were used: macroscopic and mesoscopic Hall bars. The mesoscopic sample is a Hall bar device with two current and seven voltage probes. The bar has a width W of 3.2 μm and three consecutive segments of different lengths L (2.8, 8.6, 33 μm). The macroscopic bar has a width W of 50 μm and three consecutive segments of different lengths L (100, 250 and 100 μm).

A dielectric layer was deposited (100 nm of ${\mathrm{SiO}}_{2}$ and 100 nm of ${\mathrm{Si}}_{3}{\mathrm{Ni}}_{4}$) on the sample surface and then covered by a TiAu gate. The density variation with gate voltage was $1\times 10^{11}{\;\mathrm{cm}}^{-2}V^{-1}$. The magnetotransport measurements were performed in the temperature range 1.4–4.2 K using a standard four-point circuit with a 1–27 Hz AC current of 1–10 nA through the sample, which is sufficiently low to avoid overheating effects. Six devices from the different wafers were measured, all with similar results.

## 3. Results

The variation of resistivity with gate voltage and lattice temperature for 6.4 nm quantum wells for mesoscopic and macroscopic devices is shown in Figure 1b. The current flows between contacts 1 and 5; voltage is measured between probes 2 and 3 ($\rho_{xx}=\frac{W}{L}R_{xx},\;R_{xx}=R_{1,5}^{2,3}=V_{2,3}/I_{1,5}$) for mesoscopic device; and current is applied between contacts 1 and 6; voltage is measured between probes 2 and 3 $R_{xx}=R_{1,6}^{2,3}=V_{2,3}/I_{1,6}$ for macroscopic device. The resistance behavior in zero magnetic field resembles behavior in other HgTe-based quantum wells, including topological insulators [10,12,13]: resistance shows a peak around the charge neutrality point (CNP). In graphene and zero gap HgTe wells, the CNP is coincident with the Dirac point. Transport in HgTe QWs of a critical width DC is expected to be determined by the energy gap fluctuations leading to the formation of the topological channel network [26,27]. The minimum conductivity at the Dirac point $\sigma_{xx}=1/\rho_{xx}=\frac{e^{2}}{h}\left(2.5\pm 1\right)$ agrees with the observations [27].

In the presence of a magnetic field, the energy spectrum is organized in Landau levels (LLs) with energy given by $\epsilon_{\alpha ,n}=\alpha v_{F}\sqrt{2e\hbar Bn\;}$, where α = ±1, $v_{F}$ is the Fermi velocity and n is the Landau index. Moreover, there is an additional zero energy LL, originated from Berry phase carried by each Dirac point, similar to graphene [28]. It is worth noting that, in HgTe quantum wells, Dirac fermions have a single cone (one valley) spectrum, which allows the realization of edge state transport in a strong magnetic field via counter propagating modes [13], while graphene transport depends on which degeneracy, spin or valley, is removed first in a strong magnetic field [29]. A symmetric LL spectrum around zero energy level is expected for low energy.

In this section, we present the results for the mesoscopic sample. Longitudinal $R_{xx}$ resistance has been measured as a function of gate voltage ($V_{g}$) and magnetic field (B). Figure 2 shows the the resistance color plots as a function of carrier density and B. One can see stripes corresponding to resistance maxima and minima in the B, and the slopes of the stripes are determined by the LL filling factor *ν*: $dN_{s}/dB=\nu e/h$, where h is the Plank constant.

The Dirac point corresponds to the charge neutrality point (CNP), where the Hall resistance passes zero value and changes sign [13]. The zero energy Landau level occurs at CNP splitting, due to Zeeman energy at the high magnetic field, which leads to the formation of two counter propagating states (insert in Figure 2a). Simultaneous observations of the resistance plateau in local and in nonlocal (not shown) transport confirms this scenario [13]. The experimental consequences expected for the ballistic edge transport resulting from the helical states is resistance quantization with the universal value $\frac{h}{2e^{2}}$.

Figure 2b shows the resistance trace corresponding to the chemical potential position at the CNP. One can see that the resistance plateau reaches the quantized value $\frac{h}{2e^{2}}$ in the range of the magnetic field 0.5 T<B<2 T, diverging towards the insulating value at higher B. The resistance quantization is not perfect and demonstrates mesoscopic fluctuations similar to resistance fluctuations observed in 2D topological insulators in zero magnetic field [10,12]. As was mentioned above, the mechanism of resistance deviations in TI is still under discussion [18]. In the presence of a strong magnetic field and spin–orbit interaction, backscattering between different spin-polarized chiral edge channels may occur [30].

Adapting this model for the helical edge states and assuming scattering by the Coulomb impurities in the presence of the spin–orbit coupling, we can obtain the equation for inverse scattering length [31].
(1)$$\frac{1}{l}=\frac{{\left(2\pi \right)}^{\frac{3}{2}}}{v_{1}v_{2}}{\left(\frac{e^{2}}{\hbar \varepsilon }\right)}^{2}\frac{N}{q_{s}^{2}\lambda }{\left[\frac{m\delta v\alpha g\mu H}{\delta E^{2}}\right]}^{2}$$
where $v_{1,2}$ are the velocities of the spin polarized edge states, $\delta v=v_{2}-v_{1}$, N is the impurity density, $q_{s}$ is the inverse screening length, $\lambda =\sqrt{\frac{\hbar c}{eB}}$ is the magnetic length and gμH is the Zeeman term. The energy splitting between edge states is determined by equation
(2)$$\delta E=\sqrt{{\left(g\mu H/2\right)}^{2}+{\left(mv\alpha \right)}^{2}}$$
where v is the averaged edge state velocity, α is spin orbit coupling constant and m is the effective mass. Assuming $\delta v\;\approx \frac{\delta E}{\hbar \omega_{c}}\ll v$, where $\omega_{c}=\frac{eB}{mc}$ is the cyclotron frequency, we calculate the scattering length for our system. Figure 2b demonstrates the magnetic field dependence of the scattering length for parameters: $N=10^{11}\;{\mathrm{cm}}^{-2}$, $\alpha \;\approx \;10^{5}\;m/s,\;v=10^{5}\;m/s$. The resistance can be calculated from equation $R=\left(\frac{h}{2e^{2}}\right)\left(1+L/l\right)$ [12,29]. One can see that the characteristic scattering length strongly decreases with the magnetic field and becomes comparable with the distance between probes at $B_{c}\approx 2.5\;T$. Therefore, the transport regime is expected to be ballistic below $B_{c}$, and the resistance is quantized, while the resistance increases at $B>B_{c}$ because the electrons experience more scattering.

**Figure 2 nanomaterials-12-02047-f002:**
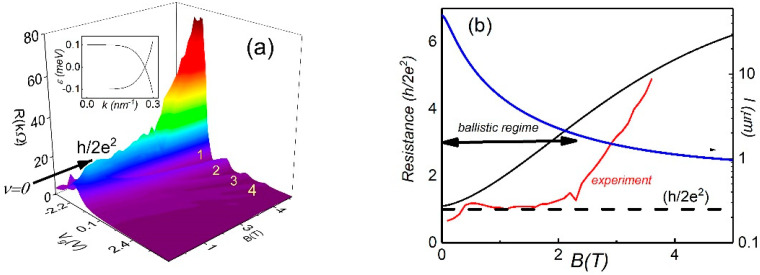
(Color online) (**a**) The color map of $R_{xx}\left(N_{s},\;B\right)$ versus $N_{s}$ and B at T=4.2 K for mesoscopic device. The arrow indicates the plateau of resistance at ν=0. The insert shows the counterpropagating spin-polarized edge states in the presence of a strong perpendicular magnetic field. (**b**) The red trace represents the longitudinal $R_{xx}$ resistance as a function of the magnetic field (B) at the CNP. The black line shows the theoretical resistance calculated from the model [31]. The blue line represents the B dependence of the mean free path calculated from the model [31].

## 4. Transport Measurements in Macroscopic Samples

In this section, we focus on the transport properties in large-sized macroscopic samples. Figure 3a shows the resistance color plots as a function of carrier density and B for a smaller field range. In addition, we invert the gate voltage scale in comparison with Figure 2a and demonstrate the hole-like spectrum of the LL on the right side of the voltage sweep. One can see a significant difference between microscopic and macroscopic sample behavior near the CNP: the resistance in the small sample is quantized and shows the plateau at low field, while the resistance in the large device is much larger than the value $h/2e^{2}$ and reveals oscillations.

Figure 3b shows the evolution of resistance at the CNP with the magnetic field. Note, that at the CNP Hall resistance is zero; therefore, the behavior of the transport coefficients in the quantum Hall effect regime and at ν=0 are very different. For example, when $\rho_{xy}\gg \rho_{xx},$ one can expect that $\rho_{xx}∼\sigma_{xx}$. In contrast, at ν=0, we observe $\rho_{xy}\approx 0$, and $\rho_{xx}∼1/\sigma_{xx}$. In Figure 3b, we plot conductivity versus the magnetic field. For comparison, we also plot the B-dependence of resistivity at $V_{g}=-2.5\;V$, corresponding to the quantum Hall regime of hole-like Landau levels. One can see the coincidence between the position of the conductivity peaks at ν=0 and the resistivity (or conductivity, because $\rho_{xx}∼\sigma_{xx}$) peaks of 2D Dirac-like holes.

To obtain more insight into the physics of the observed resistivity oscillations, it is important to consider the energy spectrum of the gapless HgTe quantum well. The particle energy in 6.4 nm HgTe wells represents a single valley cone and, aside from the Dirac-like holes in the center of the Brillion zone, the valance band contains local a valley formed by the heavy holes with a parabolic spectrum. Therefore, one can expect that LL energy in the presence of the magnetic field is asymmetric for electrons and holes [21,23,24,25].

A previous study of the quantum Hall effect in HgTe wells demonstrated strong asymmetry between electrons and holes, which was attributed to the presence of a band maximum in the spectrum of the holes [22]. It has been found that the quantized Hall plateaux for hole-like particles occurs in magnetic field three-times smaller than for electron-like carriers, and the plateau for holes is much wider that for electrons. The authors attributed such anomalous behavior to the existence of sideband holes, which may serve as reservoir and pin the Fermi level in the gap between the Landau levels of the Dirac holes. Recently heavy hole density of the states has been measured by the capacitance technique [32].

Figure 4a shows a two-dimensional color plot of the hole part of the spectrum. One can see that the resistance reveals a strikingly rich ring-like structure. Instead of stripes, expected in a conventional LL diagram, sharp abrupt bends occur at low magnetic field. The empirical linear fit is shown as dashed lines in Figure 3a and basically corresponds to the Dirac-like hole LL. The slopes of the lines decrease with the magnetic field; however, it is always larger than $\nu e/h=2.4\times 10^{10\;}{\mathrm{cm}}^{-2}/T$. For example, one can see that the slope of the first LL is close to $40\times 10^{10\;}{\mathrm{cm}}^{-2}/T$, which is about 17-times larger than expected.

It is worth noting that this unconventional LL pattern has never been observed before in other 2D systems. Indeed, in the presence of two subbands, the LL diagram shows a ring-like structure due to LL crossing [33,34,35,36] with topology, which is different from our observations. The LL crossing points become crossing two-fold owing to the crossing between spin-split first and second subbands. Instead of a diamond structure, expected from a naive picture, nonmonotonic behavior of electrochemical potential leads to a ring-like shape, although the electron–electron interactions may play a significant role too.

The LL spectrum in the valence band of HgTe becomes complicated at high energy and LL crossing occurs. We use all relevant Kane–Hamiltonian parameters to numerically calculate the density of the states for the valence band as a function of $P_{s}$ and B. The magnetic field and density coordinates of the LL crossing in the plot of the density of the states correspond to the energy level crossing, and comparison with the experiment allows to determine the Kane–Hamiltonian parameters. However, one can see that the calculations show the LL crossing at high B and density, and therefore, it is clearly insufficient to explain the ring-like pattern at low B and $P_{s}$ obtained in the experiment.

The features observed in our experiment can be understood from the consideration of the behavior of the chemical potential μ (the Fermi energy at T=0) in the presence of the reservoir formed by the density of the states originated from the tails of high index valence band LLs. To account for the key features of the model, it is important to obtain an idea of how the energy spectrum is quantized in the magnetic field.

Figure 5 shows the calculated Landau level originated from the valence band for two fixed magnetic fields. The low index levels rapidly go up with increasing B. In contrast, the high index levels form a dense set, especially near the band extreme, and are slowly shifted with increasing magnetic field. One can see that the number of levels near the maximum inside of the energy interval ΔE≈5 meV is close to 80 at B=0.5 T. As the magnetic field increases further, the level number near the maximum decreases.

The behavior of the Fermi level in a two-dimensional system strongly depends on the density of the states. In conventional 2D electron gas, the Fermi level is proportional to the charge concentration because the density of the states is constant, while in the system with a linear Dirac-like spectrum, $E_{F}$ is proportional to the square root of $N_{s}$. In the magnetic field, $E_{F}$ jumps from one level to the next lower level.

Deep minima in the diagonal resistance accompanied by a plateau in $R_{xy}$ are attributed to the existence of localized electronic states on the tail of the broadened LL in the presence of the disorder and pinning of the Fermi level. Due to the big density of LL, shown in Figure 4, the Fermi level becomes locked near $E_{c}\approx -15\;\mathrm{meV}$ (the energy at the CNP corresponds to E=−30 meV), indicated by the red line, and a further increase of density results in the overlap between heavy holes and Dirac-like holes. Note that the energy $E_{c}$ corresponds to the very low density $P_{s}=0.15\times 10^{11\;}{\mathrm{cm}}^{-2}$.

To calculate the color map plot of the density of the states D as a function of density and B, we adapted the Lorentzian form of the density of the states in a strong magnetic field [34]. Figure 4b shows the color map $D\left(N_{s},\;B\right)$, assuming level broadening independent of the magnetic field. One can see two parts of the spectrum: the low density part consists of the stripes with the large slope, corresponding to the Dirac-like hole LL at $\mu <E_{c}$, and the high density part corresponds to the overlap between Dirac-like and heavy holes with the parabolic spectrum at $\mu >E_{c}$. We also plot the slope of the LL for Dirac-like holes at high densities, corresponding to the region where $\mu >E_{c}$.

Now, we turn to a detailed comparison between the experimental resistance plot of $R_{xx}\left(P_{s},\;B\right)$ and the theoretical DOS for the LL spectrum. In the experimental fan chart, we don’t see the slope for the Dirac-like hole Landau levels at $\mu <E_{c}$ because of the broadening of the zero LL. We can resolve the LL only in a very narrow energy (density) window $0.2\times 10^{11}{\;\mathrm{cm}}^{-2}<P_{s}<1\times 10^{11}\;{\mathrm{cm}}^{-2}$. For higher densities, LL broadening abruptly increases and particle motion becomes not quantized into discrete levels.

Our model is much too simple to adequately describe the slope for the Dirac-like hole Landau levels for all densities and more advanced theory is required to describe this behavior, which is out of the scope of our experimental paper. However, the model can qualitatively explain the difference of the LL slope from the one expected. As we demonstrated above, conductivity at the CNP reveals the oscillations, which coincide with conductivity oscillations close in proximity to the CNP (Figure 3b).

We attribute this effect to the resonance scattering between helical edge states at ν=0 to the bulk LL. In narrow band gap materials, such as HgTe, potential fluctuations due to nonuniform doping play a significant role. Such potential fluctuations lead to the formation of conducting large size puddles or lakes in the bulk of the insulator, and carriers at the edge states interact with these puddles [13,16]. A number of puddles should be present in the vicinity of the edge to allow for scattering between the counter-propagating states and the bulk LL localized in each lake.

## 5. Conclusions

In this paper, we presented a detailed study of the transport in single cone Dirac fermions in 6.3–6.4 nm HgTe quantum wells in mesoscopic and macroscopic devices. We observed quantized four-terminal resistance in mesoscopic devices, which provided a stark indicator for helical edge transport at ν=0 in the presence of a magnetic field. In macroscopic samples, we observed resistance oscillations at ν=0 and an unconventional LL diagram for hole Dirac particles with several ring-like patterns.

We attribute the fan chart to LL crossing of single LL and manifold-degenerate subband levels. We reported a model considering the reservoir of the sideband hole states. The model reproduced some of the key features of the data, in particular, the density dependence of the hole LL and manifold LL crossing points. The oscillations of resistance at the CNP may occur due to the elastic intersubband scattering between the edge ν=0 state and bulk ν≠0 hole LL localized in the large size puddles near the edge.

## Figures and Tables

**Figure 1 nanomaterials-12-02047-f001:**
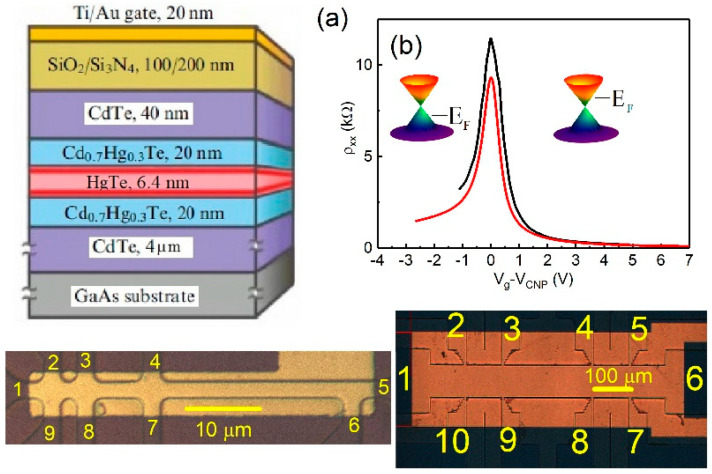
(Color online) (**a**) Schematic of the transistor. (**b**) Resistivity $\rho_{xx}$ as a function of gate voltage measured for different devices. The red trace—macroscopic and black line—mesoscopic devices. The bottom of the figure presents a top view of the samples.

**Figure 3 nanomaterials-12-02047-f003:**
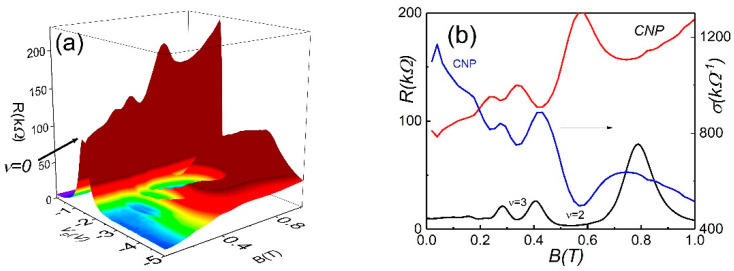
(Color online) (**a**) Color map of $R_{xx}\left(N_{s},\;B\right)$ versus $N_{s}$ and B at T=4.2 K for a macroscopic device. (**b**) The red trace represents the longitudinal $R_{xx}$ resistance as a function the magnetic field (B) at the CNP. The blue trace represents the conductivity $\sigma_{xx}$ as a function the magnetic field (B) at the CNP. The black trace represents $R_{xx}$ in the quantum Hall effect regime for 2D Dirac holes as a function the magnetic field (B) at $V_{g}=-2.5\;V$.

**Figure 4 nanomaterials-12-02047-f004:**
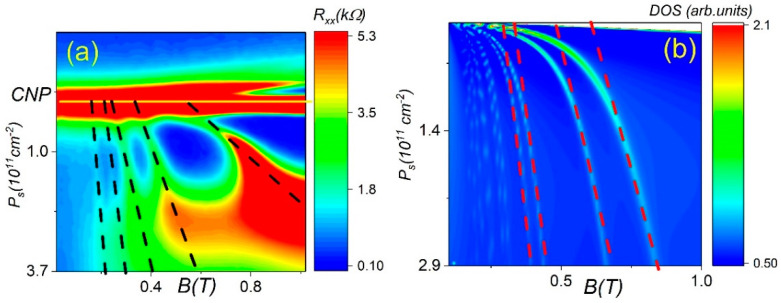
(Color online) (**a**) Color map of $R_{xx}\left(N_{s},\;B\right)$ versus $N_{s}$ and B at T=4.2 K for a macroscopic device. Black dashes represent the slope of the LL for Dirac-like holes. (**b**) Theoretical calculations of the density of states as a function of the hole density and magnetic field. Red dashes represent the slope of the LL for Dirac-like holes.

**Figure 5 nanomaterials-12-02047-f005:**
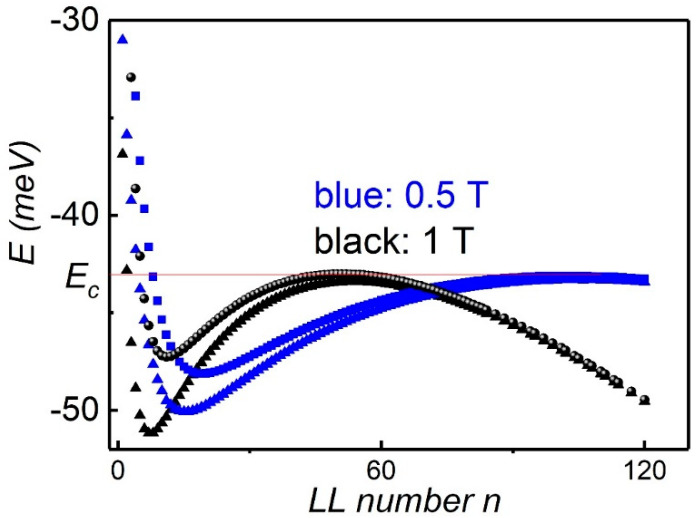
(Color online) Calculated Landau levels for a 6.4 nm symmetric HgTe quantum well for B=0.5 T and B=1 T. Two sets of levels originating from spin splitting of 2D subbands are shown. Horizontal lines show the energy when the fermi level is pinned by the backside hole LLs.

## Data Availability

The data presented in this study are available on request from the corresponding author.

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
