# Peer review of "Quantum Transport of Dirac Fermions in HgTe Gapless Quantum Wells"

_nanomaterials, 2022, doi:10.3390/nano12122047_

Round 1

Reviewer 1 Report

In the paper by Gusev et al., an experimental study of the transport properties of a single cone Dirac fermions in HgTe quantum wells is presented.  Both mesoscopic and macroscopic have been studied.  A clear indication for helical edge transport at ? = 0 in the presence of a magnetic field is observed in mesoscopic samples.  Resistance oscillations at ? = 0 and an unconventional diagram of Landau Levels are observed for hole Dirac particles in macroscopic samples.  The authors suggest a model based on crossing of a single Landau level and manifold-degenerate subband levels that adequately describes the results.  The paper is well written; the results are timely and interesting.  It should be published in its present form.

Author Response

Referee A

We thank the Referee for her/his high evaluation of our work.

Reviewer 2 Report

this work presents some interesting result on 2D topological insulator HgTe quantum well. It's good that both mesoscopic and marcroscopic samples are studied. I only have minor suggestions for the authors: please include high resolution figures with larger font size so the figure labels/axis are legible and also check for typos (I saw quite a few typos, e.g. in 6.3 - 6.4 nm the symbol "-" is typed as the division symbol throughout the paper; similar mistakes happen in a few other instances; on page 2, width of 3,2um should be 3.2 um.) 

Author Response

Referee B

We thank the Referee for her/his high evaluation of our work.

The referee wrote:

please include high resolution figures with larger font size so the figure labels/axis are legible

Our reply:

We made the corresponding correction

The referee wrote:

check for typos (I saw quite a few typos, e.g. in 6.3 - 6.4 nm the symbol "-" is typed as the division symbol throughout the paper; similar mistakes happen in a few other instances; on page 2, width of 3,2um should be 3.2 um.) 

Our reply:

We made the corresponding correction

Author Response

Referee C

We thank the Referee for her/his high evaluation of our work.

The referee wrote:

1) Line 28: …topological insulator (Quantum Spin Hall effect) …

Our reply:

We made the corresponding correction

The referee wrote:

2) How ‘zero’ is the zero‐gap material? If the authors did not do any measurement of the thickness, the may construct a plot as shown in Figs. 2a/b in Ref. 21

Our reply:

The thickness was measured via Transmission electron microscopy (TEM) of cross sectional sample and was presented in the figure 2 of the ref.27. Indeed this paper was cited (line 86-87).

The referee wrote:

3) Line 88,89. Please explain briefly why the minimum conductivity at the Dirac point is given by the formula indicated.

Our reply:

This subject has been studied in our previous publication (ref.27), where the network model is described in details. The minimum conductivity is predicted by this model and this value agrees with  our observations.

The referee wrote:

4) Re‐write the paragraph between line 90 to 98 and give it a better structure. Please mention the proportionalities in formulas.  

Our reply:

We changed the text to   formula.

The referee wrote:

5) Please improve the presentation of figure 2. Units are difficult to read and in part b), it seems like the x‐axis has been squeezed.

Our reply:

We made the corresponding correction and improve figure 2.

The referee wrote:

6) Line 140, 141: there is an issue with the absolute value and the unit of the spin‐orbit coupling constant. Please check. I assume the unit is eVm and the absolute value is incorrect.

Our reply:

\alpha has a units m/s, therefore term mv\alpha  has units mv^2, i.e energy units. We used formulas from ref. 31 ( see also supplementary materials of ref.31, eq S3 and S4). However, we  made corrections in eq.1 (line 136).  Calculations were correct and not changed.

The referee wrote:

7) Indicate the filling factors for ‐2.5 V in Figure 3.

Our reply:

We made the corresponding correction and improve figure 3.

The referee wrote:

8) Why do the authors show the conductivity and resistivity at the CNP in Figure 3 b? It is just an inverted curve.

Our reply:

As we described in the text (lines 159-167),  at CNP (  we observe , and , while in quantum Hall regime    .

In order to compare position of the peak, we inverted  curve.

The referee wrote:

9) Correct figure 4a), speed mode still on.

Our reply:

We made the corresponding correction.

The referee wrote:

10) Please use different symbols in figure 5.

Our reply:

We made the corresponding correction.

The referee wrote:

Finally, I was wondering about ‘sample statistics’ or measurements of the resistance as a function of magnetic field at different gate voltages or as a function of gate voltage using different contact pairs. Can the authors show or provide additional experimental data to make sure that the results are not sample‐dependent?

Our reply:

We have measured 2 different mesoscopic samples and dozen of the macroscopic devices. Statistic of the samples has been presented in the ref 27. Some of the results for other devices has been published in ref. 25. Indeed the measurements from the different probes and gate voltages has been performed. In order to keep the quality of the presentation and prevent confusion we decide to demonstrate the results for two representative samples.